# Predictive Scores for Late-Onset Neonatal Sepsis as an Early Diagnostic and Antimicrobial Stewardship Tool: What Have We Done So Far?

**DOI:** 10.3390/antibiotics11070928

**Published:** 2022-07-10

**Authors:** Georgia Anna Sofouli, Aimilia Kanellopoulou, Aggeliki Vervenioti, Gabriel Dimitriou, Despoina Gkentzi

**Affiliations:** Department of Paediatrics, Patras Medical School, University of Patras, 26504 Patras, Greece; gewrgianna-s@hotmail.com (G.A.S.); aim.kanellop@gmail.com (A.K.); aggelikivervenioti@gmail.com (A.V.); gdim@upatras.gr (G.D.)

**Keywords:** neonates, late-onset, LOS, sepsis, septicemia, diagnosis, prediction, score

## Abstract

Background: Late-onset neonatal sepsis (LOS) represents a significant cause of morbidity and mortality worldwide, and early diagnosis remains a challenge. Various ‘sepsis scores’ have been developed to improve early identification. The aim of the current review is to summarize the current knowledge on the utility of predictive scores in LOS as a tool for early sepsis recognition, as well as an antimicrobial stewardship tool. Methods: The following research question was developed: Can we diagnose LOS with accuracy in neonates using a predictive score? A systematic search was performed in the PubMed database from 1982 (first predictive score published) to December 2021. Results: Some (1352) articles were identified—out of which, 16 were included in the review. Eight were original scores, five were validations of already existing scores and two were mixed. Predictive models were developed by combining a variety of clinical, laboratory and other variables. The majority were found to assist in early diagnosis, but almost all had a limited diagnostic accuracy. Conclusions: There is an increasing need worldwide for a simple and accurate score to promptly predict LOS. Combinations of the selected parameters may be helpful, but until now, a single score has not been proven to be comprehensive.

## 1. Introduction

The term ‘neonatal sepsis’ (NS) is used to describe the systemic condition caused by bacteria, viruses or fungi in the first four weeks of life and is a clinical syndrome that may include signs of systemic infection, circulatory shock and multisystem organ failure [1]. Depending on the age of onset and timing of the sepsis episode, NS has been classified as either early-onset (EOS) (<72 h of life) or late-onset (LOS) (>72 h of life) [2].

Neonatal sepsis remains one of the most common causes of neonatal morbidity and mortality in developed and developing countries [3,4]. Widespread antimicrobial use for neonatal sepsis is associated with the development of resistant microorganisms, as well as adverse clinical outcomes [5,6,7]. It is estimated that 31% of global annual neonatal deaths related to sepsis could be attributed to antimicrobial resistance [8]. Simultaneously, the inability of neonates, and especially the premature ones, to moderate an inflammatory response makes them more susceptible to infectious diseases than older children or adults [9]. Notwithstanding major advances in neonatal care and increasing research, the early diagnosis of neonatal septicemia is a vexing problem and remains a great challenge to pediatricians due to multiple reasons [10]. First of all, the clinical picture is nonspecific [11,12], the signs of NS may be absent or minimal and hard to detect [13] and many noninfectious syndromes have initial clinical presentations similar to severe infections [14]. Undoubtedly, the gold standard for the diagnosis of a systemic infection is the isolation of pathogens from the peripheral blood, urine, cerebrospinal fluid (CSF) or any other sterile tissues, but unfortunately, this method is time-consuming, and its sensitivity is low [15,16]. Blood cultures may be false positives (due to contamination from the skin) or false negatives (low volume of blood in neonates, previous antimicrobial therapy, etc.) [17,18]. It is worth mentioning at this point that, if antimicrobials have been previously used, specific blood culture bottles containing resin to neutralize antibiotics (e.g., BBL^TM^ SEPTI-CHEK^TM^) should be used to prevent false negative results [17,18,19]. In addition, the rapid detection of certain pathogens such as *Streptococcus agalactiae* using the latex agglutination test (LAT) method may be suitable for use with either cerebrospinal fluid or blood samples [17,18,19]. Despite the above, the accurate diagnosis of NS remains challenging. Therefore, the term ’clinical sepsis’ (clinical features with negative cultures) is a well-recognized entity by practicing clinicians and may be more commonly encountered than positive-culture sepsis [17,18]. Furthermore, another reason that makes the detection of neonatal sepsis problematic is the relative diagnostic inaccuracy of the available parameters or biomarkers [19,20,21,22]. Procalcitonin appears to be more promising in the field, with a higher sensitivity compared to the C-reactive protein (CRP) [20,22].

As NS can be rapidly progressive and a timely diagnosis is critical [23], all the above-mentioned diagnostic difficulties steered investigators to develop the so-called ‘sepsis scores’ by combining different physical examination findings, laboratory assessments and other variables. From 1974 to 1979, U. Töllner made the first attempt to improve the early diagnosis of septicemia in newborns by the use of a score, which was published in 1982 [24]. In this first study, clinical and hematological findings of newborns with NS were studied retrospectively from the viewpoint of creating a score that was then validated prospectively. Since then, numerous studies have been published, reflecting the progressing interest in the field and the need for an accurate scoring system. Despite the remarkable efforts, a single score has not as yet been developed due to various barriers, such as an accurate and unanimous definition of neonatal sepsis, availability of laboratory tests and biomarkers in different resource settings and applicability and validation in different neonatal populations [9,10,12,13,15]. The aim of the current study is to assess the diagnostic value of predictive scores for LOS as a tool for early sepsis recognition, as well as an antimicrobial stewardship tool.

## 2. Materials and Methods

### 2.1. Information Sources

This review was conducted in accordance with the Preferred Reporting Items for Systematic Reviews and Meta-Analyses guidelines (PRISMA) [25,26]. We performed a systematic search of all available literature in the PubMed database from 1982 to December 2021.

### 2.2. Search Strategy

The following research question was developed: Can we diagnose late-onset sepsis with accuracy in neonates using a predictive score?

The following search terms were used in the Title and/or Abstract fields in various combinations: neonate(s), neonatal, newborn(s), infant(s), predict, predicting, prediction, predictive, predictor, diagnose, diagnosing, diagnosis, diagnostic, identify, identifying, identification, score(s), scoring, system(s), model(s), algorithm(s), algorithmic, calculator(s), tool(s) sepsis, septic, septicemia, septicemic, septicemia, septicemic and bloodstream infection.

### 2.3. Inclusion and Eligibility Criteria

We limited our search results to articles written in English and articles published from 1982 onwards, because the first predictive score (Töllner et al.) was published in 1982. No other limits were applied. All publications were eligible for review, with particular emphasis on research and observational studies.

### 2.4. Selection Process

We screened 1352 articles for relevance. We emphasized studies creating or validating a predictive score for the diagnosis of LOS. Therefore, we excluded studies on EOS (<72 h). Furthermore, we set the presence of a score as a perquisite for study inclusion. As a result, we excluded articles with models or algorithms that provide a prediction for LOS but not a score or a stratification of possibility for LOS. References from the screened articles were further reviewed for additional articles. Inferentially, 16 articles were included in this review, as they met all the above-mentioned criteria (Figure 1).

## 3. Results

For this systematic review, 1352 articles were identified, 33 were retrieved and 16 were included. Of them, fifteen were original studies, and one was a review/meta-analysis that included nine studies focusing predominantly on clinical parameters that might predict LOS [14]. Of the 16 studies included in the present review, 5 were conducted in India; 2 in Belgium; 2 in Thailand and 1 from each of the following countries: the USA, Germany, Australia, Canada, Turkey and Bangladesh. The review/meta-analysis was conducted in Belgium. Nine were prospective, four retrospective and two mixed (retrospective and prospective). Eight studies referred to original scores (new scores), five were validations of already existing scores and two were mixed (new scores and validations of previous scores). A total of 2664 neonates were included in the above studies.

To facilitate the presentation of the studies included in this review, we categorized them according to the type of variables used for each score. For example, if a predictive model used solely clinical parameters to predict LOS, this was categorized as clinical. On the other hand, if a predictive model used only laboratory parameters to predict LOS, this was categorized as laboratory. If a combination of parameters was used, the model was classified in the clinical/laboratory group, etc. We therefore created six different categories as follows:ClinicalLaboratoryClinical and laboratoryRisk factorsClinical, laboratory and risk factorsClinical, laboratory and management

A summary of the data from the included articles is shown in Table 1 (scores with exclusively clinical variables), Table 2 (scores with exclusively laboratory variables) and Table 3 (scores with combined variables). The diagnostic accuracy of each score is presented in Table 4.

### 3.1. Scores with Clinical Variables

The predictive scores based on Clinical Variables are shown in Table 1. First of all, in 2003, Singh et al. created the first scoring system based exclusively on clinical variables [27]. This score was validated twice; the first validation took place in 2008 by Kudawla et al. (same team) [29]. Investigators also attempted to estimate if a repetition of the assessment at different time periods would be useful. The clinical score was calculated at 0 h and at 24 h after the onset of the illness. An additional validation was held in 2010 by Rosenberg et al., with the exception that the signs of chest retraction and pre-feed aspirates were replaced by respiratory distress and poor feeding, respectively [30]. The team also sought to create a new score. The study ended up to the first bedside clinical score for nosocomial sepsis in preterm neonates, mainly addressed as low-resource settings. Therefore, the score originally created by Singh et al. was further validated prospectively, which may increase any future diagnostic utility. Few years later, in 2005, Dalgic et al. presented a comparative study in which they contrasted the Nosocomial Sepsis Predictive Score (NOSEP) (clinical, laboratory and risk factors) with a clinical score made by their NICU [28]. All patients were evaluated for sepsis both by the NOSEP score and the team’s clinical score. This was also a very useful approach, as it directly compared a score with actual clinical practice.

### 3.2. Scores with Laboratory Variables

The predictive scores based on Laboratory Variables are shown in Table 2. First of all, in 1988, Rodwell et al. created the first hematologic scoring system (HSS) for neonatal sepsis (both EOS and LOS), consisting exclusively of laboratory variables [23]. The HSS was then prospectively evaluated in 2011 by Narasimha and Harendra and again in 2013 by Makkar et al. [31,32]. Of note, the use of laboratory values in the score created by Rodwell et al. was found to have high sensitivity but low specificity in LOS sepsis diagnosis (Table 4).

### 3.3. Scores with Clinical and Laboratory Variables

The predictive scores based on both Clinical and Laboratory Variables are shown in Table 3. In 1982, U. Töllner created the first predictive score for the early diagnosis of septicemia in newborns [24]. The analysis was divided into three phases: symptoms before (when the patient showed no changes in their clinical and hematological values), at the beginning (upon the initial presentation of symptoms of septicemia or hematological changes) and at the peak of the illness (with all clinical symptoms of septicemia/septic shock present). The score was also tested in a prospective cohort on not only septic and healthy neonates but also on neonates with other clinical conditions. This score was, at that point, the most comprehensive one in terms of variable inclusion and set the background for the development of further scores.

A more detailed approach was made by Griffin et al. in 2007, where the Heart Rate Characteristics (HRC) were analyzed in addition to other known clinical and laboratory findings [33]. They also created a score containing variables connected to NS. The researchers recorded signs and symptoms before, at the time and after the BC. The calculated HRC index adjunctive to the clinical information was found to be useful in LOS predictions.

Finally, the first predictive model for bacterial LOS was presented by Husada et al. in 2020, incorporating a variety of parameters, and was found to have high sensitivity, as well as specificity (Table 4) [34].

### 3.4. Scores Based on Risk Factors

In 1994, Singh et al. created the only scoring system for the prediction of NS, using solely perinatal risk factors [39]. The investigators examined the interdependence of each variable, categorized them as dependent or independent factors and subsequently developed a score not only for EOS but LOS as well. This score was found to have high sensitivity but very low sensitivity (Table 4). No further predictive scores were created solely based on the risk factors.

### 3.5. Scores Based on Clinical, Laboratory and Risk Factors

In 2000, Mahieu et al. presented a bedside scoring system named NOSEP for NICUs, targeting particular nosocomial sepsis (occurring >48 h of admission) [35]. The NOSEP-2 score was composed by adding into the NOSEP score the culture results of central vascular catheters. An internal and external validation of the NOSEP score in six NICUs were displayed by Mahieu et al. in 2002 [36]. Additionally, in order to increase the predictive performance of the score, the investigators developed NOSEP-NEW-I and NOSEP-NEW-II scores by changing the cut-off values and including additional variables, respectively. All the above scores, shown in Table 3, were generally more comprehensive and well-validated in larger cohorts of neonates. Their sensitivity was high, and their negative predictive value was higher compared to other categories of scores (Table 4).

### 3.6. Clinical, Laboratory and Management

Few scores were based on clinical and laboratory variables as well as management decisions (Table 3). In particular, in 2005, Okascharoen et al. presented the first bedside scoring system for hospitalized neonates, after examining multiple variables associated with proven LOS [37]. In 2007, the same team presented an external validation of this scoring system [38]. Simultaneously, clinicians were asked to complete a questionnaire and rate the probability of true sepsis after obtaining basic laboratory results while they were not aware of the criteria of the LOS score. The researchers concluded that clinicians may predict LOS as accurately as the scoring system but tend to overestimate the possibility of LOS and the score performed better in prediction compared to clinicians’ judgment. This score was also found to have high sensitivity and a high negative predictive value (Table 4).

## 4. Discussion

Late-onset neonatal sepsis is one of the most challenging areas in neonatal medicine today. While it is crucial to diagnose NS early, it is equally important not to overuse antibiotics.

### 4.1. Predictive Scores: Clinical vs. Laboratory Parameters

Predictive scores are powerful tools to improve clinical decision-making; they simplify the decision-making procedure and assist the clinicians in increasing the accuracy of the diagnostic assessment [40,41]. Predictive models may facilitate medical judgment through a more evidence-based procedure. In order to make predictive scores the cornerstone of the early diagnosis of NS, we are in need of an accurate and easy-to-use model. A guide for how these prediction models should be structured was published as a protocol form [42].

Scores based solely on clinical symptoms and signs could be easily used and are of major importance in low-/middle-resource settings and in primary care, where a laboratory assessment may be inaccessible or unaffordable. Additionally, the evaluation of these scores saves time. However, a significant limitation of exclusively clinical scores is the subjectivity needed for the assessment of many clinical parameters (such as lethargy, pallor and hepatomegaly), which makes it more demanding for the inexperienced clinician.

On the other hand, scores containing exclusively laboratory variables provide a more objective and thus accurate tool for the decision-making process. The undemanding implementation by the inexperienced physician makes them a more standard basis for clinical practice. Nonetheless, laboratory data demand a requisite period of time, sometimes of vital importance, such as in emergency situations, and may be scarce in developing countries and primary care. In addition, it must be taken into consideration that hematologic response varies, according to multiple factors (gestational and postnatal age, time between onset of infection and the blood sample, etc.). It should also be underlined that, because of the rapid changes in the process of the illness, it is crucial for the score to be able to be repeated at short intervals of time.

As a result, we consider that the golden ratio for an ideal score may be the combination of clinical and laboratory variables. Clinical points will give a suspicion of LOS without waiting for the lab results (clinician can give empirical therapy), and the lab results will confirm this suspicion or not (clinician continues or withdraws treatment). Adding extra indicators such as risk factors for NS or management variables helps clinicians to keep in mind children with a higher risk for NS. Generally, the knowledge that hematologic parameters change rapidly, in comparison with clinical signs or management factors, is pivotal for the assessment of the variables when composing the score.

### 4.2. Prematurity and Low Birth Weight (LBW) Infants

Plenty of the scores were tested in preterm/very preterm and/or LBW/VLBW neonates [27,29,30,32,33,37]. Three studies pointed out that NS was more common in preterm and/or LBW patients [23,33,39]. The assessment in this population is extremely helpful, because premature and LBW patients consist of a significant part of neonates and especially groups with a higher risk for LOS. Nevertheless, these babies do not respond to sepsis as full-term infants. For example, a fever is rarer in premature babies, and hematologic responses vary according to age and BW. Additionally, symptoms such as apnea, chest retractions and grunting were more common in premature babies because of lung immaturity, and as a result, we should consider them as less appropriate to predict NS in all age neonates (signs less specific but highly sensitive). For example, the external validation of Singh et al. held by Rosenberg et al. in 2010 performed significantly lower than the original study, and this fact was explained by the investigators due to the differences in the natures of the population; the original score was addressed to all neonates admitted to the NICU (although it gave emphasis on preterm and LBW), while the validation study involved only very preterm neonates (≤33 weeks gestational age, admitted in NICU under 72 h of life), which are generally more prone to respiratory symptoms because of a more severe lung immaturity. Moreover, in the study of Okascharoen et al. in 2005, in the validation set, there was a significant increase in the amount of preterm (from 40% to 69%) and LBW infants (18% to 49%) compared to a derivation study. Therefore, the scores tested in both groups in tandem may leave a clouded picture, concerning their accuracy in the diagnosis of LOS. A score applicable to all newborns or at least a score composed, tested and used in a particular group (only preterm, only term, etc.) is something to be discussed.

### 4.3. Clinical and Laboratory Parameters of High Diagnostic Value

Taking into consideration the results reported in each study of our review, overall, apnea, lethargy, tachycardia, pre-feed aspirates, changes in skin coloration, abnormal temperature and abnormal HR seemed to be the most sensitive, while grunting, hypothermia and chest retractions the most specific clinical signs. Feeding intolerance, hypotonia, lethargy and fever were clinical signs highly predictive of NS. One study and its validation indicated UVC usage as a sensitive parameter [37,38]. Some studies pointed out fever as a significant symptom of NS [34,35,37]. One should bear in mind though that neonates, especially the preterm ones, cannot develop a febrile response similar to the term ones, due to a lack of immune system development. Hence, hypothermia is also a worrying clinical sign.

As for the laboratory findings, the immature to total (I:T) neutrophil count ratio, admit immature PMN count, immature to mature (I:M) neutrophil count ratio (I:M ratio) and total PMN count appeared as the most sensitive, whereas the PLT count, degenerative changes, total WBC count, I:T ratio and I:M ratio are the most specific variables. The I:T ratio, PLT count, I:M ratio and neutrophil fraction were found to be highly predictive laboratory signs. Three studies indicated that the I:T ratio may be the most reliable laboratory sign for prediction [31,32,33].

### 4.4. New Diagnostic Techniques

Besides the above-mentioned and discussed scores, new diagnostic techniques try to approach the challenge of the accurate and early diagnosis of NS. Machine learning is a subfield of artificial intelligence and uses particular electronic data to train and validate artificial Neural Networks in order to create diagnostic models and algorithms. A great number of data (signs, symptoms and especially monitoring and laboratory data) are gathered by septic and non-septic infants and assemble a computer-based algorithm that, based on precise coefficients of the variables, can diagnose or predict NS. The outcome of these models can be sepsis/no sepsis or/and a stratification of the possibility for NS. The main purpose of this approach is to introduce a more personalized basis for the diagnosis of neonatal sepsis, relying on precise and continuous information. A significant number of such studies reflect the interest on this new wrinkle, with impressive results in their diagnostic power [43,44,45,46,47,48]. Undoubtedly, this idea provides a pioneering perspective on the field, not only for the time being where NICUs count on continuous monitoring, but also for the near future when computing methodologies will play a crucial role in medical decision-making generally. In our review, these studies were excluded because we did set a strict limit to studies containing arithmetical scores only. However, the preexisting data presented in this manuscript can reveal trends that are missed by cursory or even more detailed analysis. The use of high-speed computing where multiple factors are incorporated could determine the optimal combination of the clinical and laboratory-based criteria. The reported sensitivity and specificity values in Table 4, along with the positive/negative predictive values, could be used in this type of approach. Instead of a single score, a two-tiered scoring system could be developed. One that integrates the intrinsic value of each defined measurement for early screening, and the latter for a definitive diagnosis. For instance, the early score is based on high sensitivity and for screening infants for NS using combined clinical and laboratory data suggested by computer analysis. Sensitivity is given priority over specificity when screening for NS so that false negatives could be avoided. However, this will lead to an increased level of false positives. Consequently, the definitive score would be used to rule out any false positives. This score would need to be highly specific but less sensitive. This approach should make the final determination of LOS more objective and lead to appropriate antimicrobial therapy.

### 4.5. Strengths and Limitations

This study is the first systematic review of all the existing scores containing all possible variables for the early diagnosis of LOS. We acknowledge though that our study also has limitations. The main one is that different definitions were used for LOS in the studies included. There were studies in which sepsis was defined as positive blood or a CSF culture, studies that defined sepsis as two positive blood cultures for the same organism and other studies in which a positive culture combined with clinical findings suggestive of sepsis were required for definition. In addition, as LOS definitions requires culture proven infection the true disease incidence might have been underestimated which in turn may limit the diagnostic accuracy of the scores. Moreover, we performed our search on the PubMed database, and hence, we might have missed some scores published in other databases. Finally, three of the scores in our review included few data on EOS; hence, their diagnostic ability may not be as accurate for LOS compared to the rest of the studies that have included only LOS cases in their predictive models.

## 5. Conclusions

Predictive scores for late-onset neonatal sepsis have the potential to represent a useful tool for early diagnosis and for guiding whether an individual patient needs antimicrobials. Inferentially, a future goal is to find the golden ratio between objective clinical, basic laboratory and other pivotal variables for composing the ideal score so as to improve the diagnostic accuracy and rationalize the antibiotic use. Lessons learnt from the studies until now will be vital for the introduction of new diagnostic scores not only for NICUs and Emergency Centers but also for low-source settings. So far, as we do not have the congruous score or model, we must continue the efforts to determine the optimum course of action without reckoning the thesis that any prediction model should play an adjunctive and supplementary role in medical judgment and not supersede the fundamental clinical opinions.

## Figures and Tables

**Figure 1 antibiotics-11-00928-f001:**
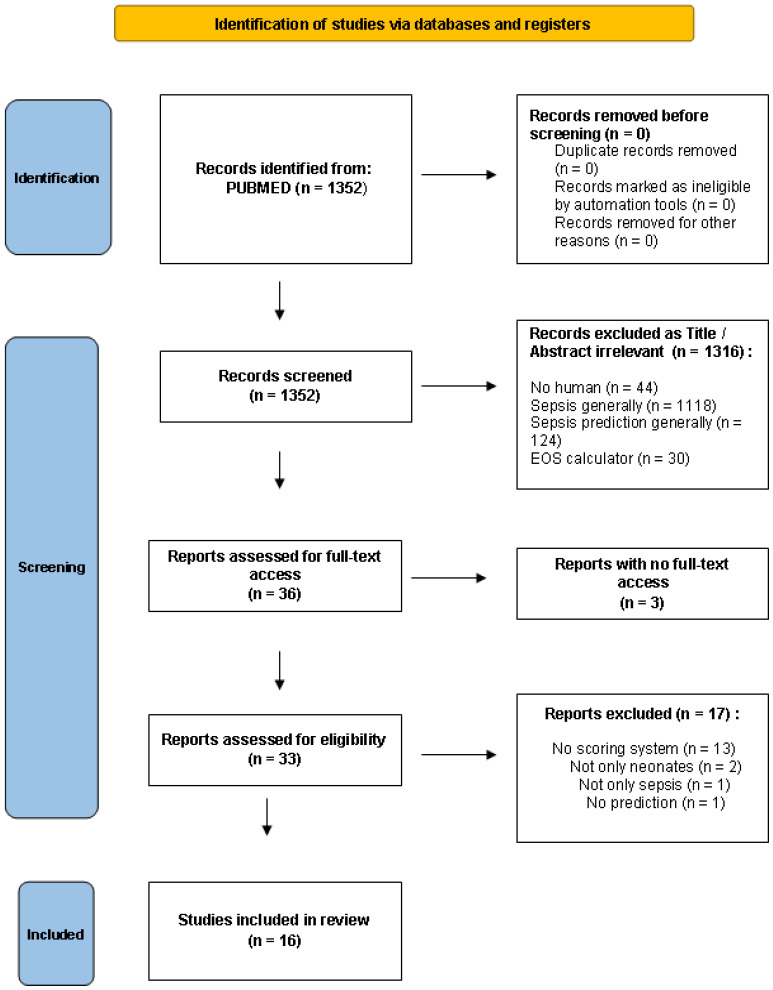
Preferred Reporting Items for Systematic Reviews and Meta-Analyses (PRISMA) 2020 diagram of the studies included at each stage of the screening process.

**Table 1 antibiotics-11-00928-t001:** Predictive scores for the LOS with clinical variables.

Reference	Country, Year	Method	Design	Population	Scoring System (Points)	Main Findings
[27]	India, 2003	Prospective	Original	80 neonates: 105 episodes (30 definite, 17 probable sepsis and 58 no sepsis) 91% preterm, 93% LBW	grunting (2) abdominal distension (2) increased pre-feed aspirate (1) tachycardia (1) hyperthermia (1) chest retraction (1) lethargy (1)	Score different in septic and no septic infants. Most prevalent signs in septic babies: apnea, lethargy, tachycardia. Most specific signs in septic babies: grunting, hypothermia, chest retractions.
[28]	Turkey, 2005	Retrospective	Original and external validation (comparison with NOSEP score of Mahieu et al.)	102 neonates: 132 episodes (51 blood culture (+), 51 no sepsis)	respiratory symptom (2) abdominal distension (2) feeding intolerance (2) hypotension (2) bradycardia (2) lowest and highest body temperature difference (2)	Score different in septic and no septic infants. Feeding intolerance and higher I:T ratio as significant predictors of NS.
[29]	India, 2008	Prospective	Validation (of Singh et al.)	202 neonates: 220 episodes (60 definite sepsis) Weight: 1000–2500 g	grunting abdominal distension increased pre-feed aspirate tachycardia hyperthermia chest retraction lethargy ≥1 = positive clinical score	The most frequent signs in septic infants: lethargy, apnea and pre feeds aspirates. All clinical signs decreased in frequency from 0 h to 24 h. Different score at 0 h and at 24 h: Se better at 0 h (all sick neonates included), Sp, PPV, NPV better at 24 h. Better prediction of NS at 24 h (PPV↑ at 24 h). Score combined with sepsis screen: ↑Se, NPV but ↓Sp, PPV
[30]	Bangladesh, 2010	Retrospective	Validation (of Singh et al.) and original	160 neonates: 193 episodes (105 culture (+) in 98 neonates, 88 culture (−) in 79 neonates)GA ≤ 33 weeks (very preterm), ≤72 h admitted to hospital	grunting (2) abdominal distension (2) poor feeding (1) tachycardia (1) hyperthermia (1) respiratory distress (1) lethargy (1)	First bedside clinical score for very premature neonates in a low-resource setting. This external validation performed significantly lower Sensitivity than the original study. As the number of sings presented within 48 h of sepsis evaluation was increased, Se and NPV were reduced, while Sp and PPV were augmented. Sensitivity reducing when more than 1 signs were present.

**Table 2 antibiotics-11-00928-t002:** Predictive scores for the LOS with laboratory variables.

Reference	Country, Year	Method	Design	Population	Scoring System (Points)	Main Findings
[23]	Australia, 1988	Prospective	Original (HSS)	287 neonates: 298 episodes (27 sepsis, 23 probable infection, 248 non infected) Group 1: 243 neonates (≤24 h of age) Subgroup 1: 113 neonates (preterm) Subgroup 2: 130 neonates (term) Group 2: 55 neonates (days 2–30) Age 1–30 days (EOS and LOS), with perinatal risk factors or clinical suspicion of sepsis	Immature to total neutrophil(I:T) ratio (↑) (1) Total PMN count (↑/↓) (1 or 2) Immature to mature (I:M) neutrophil ratio (≥0.3) (1) Immature PMN count (↑) (1) Total WBC count (↑/↓: ≤5.000/mm^3^ or ≥25.000, 30.000 and 21.000 at birth, 12–24 h and day 2 onward, respectively) (1) Degenerative changes in PMNs ≥ 3 (1), +for vacuolization, toxic granulation, or Döhle bodies PLT count (≤150.000/mm^3^) (1)	Sepsis more common in preterm than in term neonates. I:T ratio, abnormal PMN count and I:M ratio: the most frequent lab findings. Most specific sings: PLTs, degenerative changes. The higher the score, the greater the probability of NS. Cut-off score performed better than the most accurate hematologic variable (I:T ratio). HSS provides an objective assessment. Many factors can affect hematologic response. Importance in combining lab + clinical data. Emphasis on EOS rather than LOS. Suggested as a screening test for diagnosing NS.
[31]	India, 2011	Prospective	Validation of HSS (by Rodwell et al.)	50 neonates: 50 episodes (12 sepsis, 26 probable infections, 12 no sepsis) Aged 24 h–8 days (EOS and LOS), 58% term and 42% preterm, with perinatal risk factors or clinical suspicion of sepsis.	I:T ratio (↑) (1) Total PMN count (↑/↓) (1 or 2) I:M ratio (≥0.3) (1) Immature PMN count (↑) (1) Total WBC count (↑/↓: ≤5.000/mm^3^ or ≥25.000, 30.000 and 21.000 at birth, 12–24 h and day 2 onward, respectively) (1) Degenerative changes in PMNs ≥ 3 (1), +for vacuolization, toxic granulation, or Döhle bodies PLT count (≤150.000/mm^3^) (1)	Total PMN count and immature PMN count: the most sensitive signs in sepsis. Total WBC count, I:T ratio and PLT count: the most specific findings in sepsis. Best PPV: I:T ratio and PLT count. I:T ratio and degenerative changes: the most reliable variables. The higher the score, the greater the probability of NS. Suggested as a screening test for diagnosing NS.
[32]	India, 2013	Prospective	Validation of HSS (by Rodwell et al.)	110 neonates: 110 episodes (42 sepsis, 22 probable infection, 46 normal) Age birth-1 week (EOS and LOS), 57% preterm and 43% term, with perinatal risk factors or clinical suspicion of sepsis	I:T ratio (↑) (1) Total PMN count (↑/↓) (1 or 2) I:M ratio (≥0.3) (1) Immature PMN count (↑) (1) Total WBC count (↑/↓: ≤5.000/mm^3^ or ≥25.000, 30.000 and 21.000 at birth, 12–24 h and day 2 onward, respectively) (1) Degenerative changes in PMNs ≥ 3 (1), +for vacuolization, toxic granulation, or Döhle bodies PLT count (≤150.000/mm^3^) (1)	Immature PMN: the most sensitive variable. I:M ratio: the most specific and the most predictive sign. I:T ratio: the most reliable indicator of sepsis. HSS more sensitive, specific and predictive in preterm than in term neonates. The higher the score, the higher the likelihood of NS. Emphasis on preterm (57%) than in term. Suggested as screening test for diagnosing NS.

**Table 3 antibiotics-11-00928-t003:** Predictive scores for the LOS with combined variables.

Reference	Country, Year	Method	Design	Population	Scoring System (Points)	Main Findings
[24]	Germany, 1982	Retrospective and prospective	Original	403 neonates: Retrospective: 83 with sepsis Prospective: 39 with sepsis, 42 with amniotic infection, 28 with post-asphyxia syndrome, 28 premature with cerebral hemorrhage, 183 controls	skin coloration (0–4) microcirculation (0–3) metabolic acidosis (0–2) muscular hypotonia (0–2) bradycardias (0–1) apneic spells (0–1) respiratory distress (0–2) liver enlargement (0–1) gastrointestinal symptoms (0–1) WBC count (0–3) Shift to the left (0–3) thrombocytopenia (0–2)	Analysis was divided into 3 phases: onset, at the beginning and at the peak of the illness. Each phase gave different results: as the illness evolved, the scores got higher. Changes in skin coloration: the most frequent sign of NS. Septic neonates performed high scores (47% at the beginning of the illness and 92% in seriously ill infants), in contrast with non-septic neonates.
[33]	USA, 2007	Prospective	Original	337 neonates: 76 episodes of proven sepsis (blood culture (+) in 63 neonates, 80 episodes of clinical sepsis (blood culture (–) in 63 neonates Age ≥ 7 days old and ≥ 7 days of HRC monitoring Out of 337 neonates: 172 were < 1500 g (VLBW)	Feeding intolerance (2) Severe apnea (2), 50% increase in the number of apneic episodes over a 24 h period in an infant stable for 3 days (2) I:T ratio > 0.2 (2) Increase in ventilatory support and FiO2 by 25% from baseline (1) Lethargy or hypotonia (1) Temperature instability (>38 °C or <36.2 °C), 2 episodes within 8 h (1) Hyperglycemia (>180 mg/dL) (1) Abnormal WBC count (>25.000 or <5.000)	Hyperglycemia and abnormal WBC count: highly associated with NS only the time of the blood culture. Hypotonia and lethargy: great association with NS only the time preceding the blood culture. Infants with sepsis had higher scores than controls. Hypotension in only 3% of infants with NS (not included in the score). HRC index and clinical score were predictive for NS in the next 24 h. Clinical tests less useful before the NS, because signs and symptoms are present less often. Infants with clinical or proven sepsis: higher scores than controls. Feeding intolerance: the most predictive clinical sign of NS. Feeding intolerance, hypotonia, lethargy and abnormal I:T ratio: the most predictive findings. I:T ratio the most robust independent predictor. Increase in the score in the 24 h before the clinical diagnosis HRC index adjunctive to clinical information proved useful.
[34]	Thailand, 2020	Retrospective	Original	208 neonates: 52 sepsis (only proven bacterial LOS), 156 controls Aged ≥ 7 days	poor feeding (2) abnormal heart rate (outside the range 100–180 x/min) (3) abnormal temperature (outside the range 36–37.9 °C) (4) abnormal O2 saturation (<92%) (1) abnormal leukocytes (outside the range of 5.000–20.000/cm) (2) abnormal pH (outside the range of range 7.27–7.45) (2)	Duration of hospitalization, intracranial hemorrhage, high-risk pregnancies and resuscitation: the most powerful risk factors. Abnormal temperature and abnormal HR: the most common sings in NS. Abnormal HRC occurred early in the course of the illness. Abnormalities were found 12–24 h before the clinical diagnosis of NS. No infant with hypothermia had LOS. Antibiotic therapy to be guided according to the score.
[35]	Belgium, 2000	Prospective and retrospective	Original (NOSEP score)	119 neonates: 154 episodes: Derivation cohort: 104 episodes of presumed NS in 80 neonates (43 proven sepsis)Validation cohort: 50 episodes of proven NS in 39 neonates >48 h in NICU	NOSEP-1 score: Fever > 38.2 °C (5) CRP ≥ 14 mg/L (5) Thrombocytopenia < 150 × 10^9^/L (5) Neutrophil fraction > 50% (3) Total parenteral nutrition (TPN) ≥ 14 days (6) NOSEP-2 score: NOSEP-1 score + culture results	Score for nosocomial NS. BW, GA, presence of CVC, prolonged hospital stay and exposure to TPN (especially lipid emulsions) > 14 days: strongly associated with NS. TPN as the only independently associated factor. Fever and neutrophil fraction as powerful signs for prediction. Adding catheter cultures improves the diagnostic power of the score. NOSEP score as accurate as a continuous computerized scoring system. Only 2 variables do not rely on lab results. Waiting for the results for assessment.
[36]	Belgium, 2002	Prospective	Validation of NOSEP score (by Mahieu et al.) and new score	128 neonates: 155 episodes:Internal validation: 62 episodes of presumed NS in 49 neonates (20 proven NS) External validation: 93 episodes of presumed NS in 79 neonates (51 proven NS) >48 h in NICU	NOSEP-1 score: Fever > 38.2 °C (5) CRP≥ 14 mg/L (5) Thrombocytopenia < 150 × 10^9^/L (5) Neutrophil fraction > 50% (3) Total parenteral nutrition (TPN) ≥ 14 days (6) NOSEP-2 score: NOSEP-1 score + culture results NOSEP-NEW-I: Fever > 38.1 °C (5) CRP ≥ 30 mg/L (5) Thrombocytopenia < 190 × 10^9^/L (5) Neutrophil fraction > 63% (3) Total parenteral nutrition (TPN) ≥ 15 days (6) NOSEP-NEW-II: NOSEP-NEW-I + recent surgery, maternal hypertension and ventilation at time of sepsis work up.	Score for nosocomial NS. External validation was set in multiple NICUs. Score was higher in septic neonates in both internal and external validations. Internal validation was better than the external. Score suggested as a tool for detection of NS and for reduction of unnecessary use of antibiotics in NICUs.
[37]	Thailand, 2005	Retrospective	Original	173 neonates: Derivation phase: 100 neonates (17 NS), 40% premature and 18% LBW Validation phase: 73 neonates (25 NS), 69% premature and 49% LBW Hospitalized for >72 h after birth	Hypotension (4) Abnormal body temperature (>38 °C or <36.5 °C or temperature instability) (3) Respiratory insufficiency (apnea/bradycardia, tachypnea, cyanosis, increased oxygen requirement or ventilator settings) (2) Neutrophil Band form fraction ≥ 1% (2) Thrombocytopenia (<150 × 10^3^/μL) (2) Umbilical venous catheterization: 1–7 days (2), >7 days (4)	First bedside score for neonates hospitalized > 72 h. Hypotension and abnormal body temperature had the strongest association with NS. Risk variables: LBW, prematurity and TPN: no significant association with LOS, while UVC usage independently associated. Combination of clinical, laboratory and management variables: suspicion of LOS without waiting for the lab results. Score based mostly on clinical sings. Risk groups: stratification of risk for LOS (low, intermediate, high risk) and benefit for decision-making.
[38]	Canada, 2007	Prospective	Validation (of Okascharoen et al.)	105 neonates: 35 NS Aged 2–90 days >48 h in NICU	Hypotension (4) Abnormal body temperature (>38 °C or <36.5 °C or temperature instability) (3) Respiratory insufficiency (apnea/bradycardia, tachypnea, cyanosis, increased oxygen requirement or ventilator settings) (2) Neutrophil band form fraction ≥ 1% (2) Thrombocytopenia (<150 × 10^3^/μL) (2) Umbilical venous catheterization: 1–7 days (2), >7 days (4)	No significant difference in GA, BW, utilization of CVC and duration of TPN between septic and non septic children. Only utilization of UVC proved to make a difference. External validation performed similar accuracy with the internal validation. From low to intermediate risk: Se, Sp ↓ Clinicians predict LOS as strongly as the scoring system, but tend to overestimate the possibility of LOS: score performed better in prediction comparing to clinicians viewpoint. When the neonatal population consists only of proven LOS records, NS was underestimated, while when suspected LOS episodes are present, LOS tended to be overestimated.

**Table 4 antibiotics-11-00928-t004:** Diagnostic power of the predictive scores as calculated in each study.

Reference	Model Application/Cut-Off Score	Sensitivity	Specificity	Positive Predictive Value	Negative Predictive Value
[39]	≥1 for definite sepsis	87	29	38	85
	≥1 for definite +/or probable sepsis	81	29	48	65
[28]	NOSEP score (8–24)	64	58	45	75
	Clinical score (6–12)	56	71	86	33
[29]	≥1				
	0 h	90	22.5	30.3	85.7
	24 h	75	60.6	41.7	86.6
	(+) screen and/or 0 h	95	18.1	30.3	90.6
[30]	Singh et al. score	56.6	52.1	78.1	28.4
	Score: 1	77.1	50	64.9	64.7
[23]	≥3 for sepsis	96	78	31	99
	≥3 for sepsis or probable infection	98	-	58	-
[31]	≤2: sepsis is unlikely 3–4 sepsis is possible ≥5 sepsis or infection is very likely	-	-	-	-
[32]	≤2: sepsis is unlikely 3–4 sepsis is possible ≥5 sepsis or infection is very likely	-	-	-	-
[24]	<4,5: sepsis is excluded with high probability 5–10: probable infection that leads to sepsis >10: sepsis is certain	-	-	-	-
[33]		-	-	-	-
[34]	2	88.5	90.4	75.4	95.9
	3	82.7	93.6	81.1	94.2
[39]	≥1 for LBW ≥4 for NBW	-	-	-	-
[35]	NOSEP ≥ 8	95	43	54	93
[36]	NOSEP	73	57	67	63
	NOSEP-NEW-I	84	42	64	69
	NOSEP-NEW-II	82	67	75	76
[37]	Validation set: 4	92	56	56	90
[38]	3	97	39	43	96

## Data Availability

Not applicable.

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
