# Peer review of "Predictive Scores for Late-Onset Neonatal Sepsis as an Early Diagnostic and Antimicrobial Stewardship Tool: What Have We Done So Far?"

_antibiotics, 2022, doi:10.3390/antibiotics11070928_

Round 1

Reviewer 1 Report

The authors herein present an interesting review on late neonatal sepsis, highlighting how much work is still missing on this threatening condition.

The work is well written and also well conducted. The beginning of chapter 4.2 is a bit confusing and should be more straightforward.

Please check again the references, as some mistakes in format as well as citations are present.

Author Response

- The authors herein present an interesting review on late neonatal sepsis, highlighting how much work is still missing on this threatening condition. The work is well written and also well conducted.  The beginning of chapter 4.2 is a bit confusing and should be more straightforward.

- Thank you for your positive feedback on the manuscript. Following your advice we have rephrased the beginning of chapter 4.2 to avoid any confusion as following: Plenty of the scores were tested in preterm/very preterm and/or LBW/VLBW neonates

- Please check again the references, as some mistakes in format as well as citations are present.

- Thank you for the comment. We have gone through an extensive reference check and corrected formatting and citations mistakes.

Reviewer 2 Report

General comments:

The authors discuss finding a “golden ratio between objective clinical, basic laboratory, and other pivotal variables” in the Conclusion. This concept is of value and should be further developed in conjunction with information seen in section 4.3. New diagnostic techniques (Line 315). Here, the authors indicate the use of artificial intelligence (AI) to develop “computer-based algorithm” to diagnose or predict LOS. Data mining of preexisting patient data presented in this manuscript and other sources can reveal trends that are missed by cursory or even more detailed analysis. The use of high-speed computing where multiple factors are “plugged in” could determine the optimal combination (sweet spot) between clinical and laboratory-based criteria. A greater weighting should be assigned to objective criteria included in clinical signs/symptoms and laboratory data.

The reported sensitivity and specificity values in Table 4 along with positive/negative predictive values could be used in this type of approach. Instead of a single score, a two-tiered scoring system should be considered. One that integrates the intrinsic value of each defined measurement for early screening and the later for a definitive diagnosis. For example, the early score is based on high sensitivity for screening infants for NS using combined clinical and laboratory data suggested by computer analysis. Sensitivity is given priority over specificity when screening for NS so that false negatives are avoided. However, this will lead to an increased level of false positives. Consequently, the definitive score would be used to rule out any false positives. This score would need to be highly specific but less sensitive. This approach should make the final determination of LOS more objective and lead to appropriate antimicrobial therapy.

Specific comments:

Line 29 - Recommend additional characterization of neonatal sepsis indicating that it is a clinical syndrome that may include signs of systemic infection, circulatory shock, and multisystem organ failure.

Line 55 - The authors indicate microbe culture recovery as the “gold standard” but neglect mentioning rapid detection of certain neonatal sepsis agents such as Streptococcus agalactiae (group B streptococcus) using latex agglutination test (LAT) method suitable for use with either cerebrospinal fluid or blood samples.

Line 58 - If antimicrobials are likely, blood culture bottles containing resin to neutralize antibiotics (e.g., BBL™ SEPTI-CHEK™) should be routinely included in suspected cases of neonatal sepsis to prevent false negative results.

Line 63 - Consider mentioning that procalcitonin is being investigated as an acute-phase reactant marker for neonatal sepsis. It is very sensitive, when compared to C-reactive protein (CRP) but is less specific.

Line 72 - The authors should consider outlining criteria included in Töllner’s scoring method and saying why it is inadequate.  Also, state major barrier(s) to developing an objective/accurate single scoring method.  

Line 174 - “The clinical score was calculated at 0h and at 24h of the onset of the illness”. Indicate that these time points are associated with early-onset (<72hours).

Table 3 - Points (  ) missing after Abnormal WBC count (ref: Griffith et al 2007).

Table 4 - Specify source of Sensitivity/Specificity and Positive Predictive Value/Negative Predictive Value information (e.g. each study, calculated, etc.).

Line 295 - Use “full-term” instead of “terms”.

Author Response

General comments:

- The authors discuss finding a “golden ratio between objective clinical, basic laboratory, and other pivotal variables” in the Conclusion. This concept is of value and should be further developed in conjunction with information seen in section 4.3. New diagnostic techniques (Line 315). Here, the authors indicate the use of artificial intelligence (AI) to develop “computer-based algorithm” to diagnose or predict LOS. Data mining of preexisting patient data presented in this manuscript and other sources can reveal trends that are missed by cursory or even more detailed analysis. The use of high-speed computing where multiple factors are “plugged in” could determine the optimal combination (sweet spot) between clinical and laboratory-based criteria. A greater weighting should be assigned to objective criteria included in clinical signs/symptoms and laboratory data.

The reported sensitivity and specificity values in Table 4 along with positive/negative predictive values could be used in this type of approach. Instead of a single score, a two-tiered scoring system should be considered. One that integrates the intrinsic value of each defined measurement for early screening and the later for a definitive diagnosis. For example, the early score is based on high sensitivity for screening infants for NS using combined clinical and laboratory data suggested by computer analysis. Sensitivity is given priority over specificity when screening for NS so that false negatives are avoided. However, this will lead to an increased level of false positives. Consequently, the definitive score would be used to rule out any false positives. This score would need to be highly specific but less sensitive. This approach should make the final determination of LOS more objective and lead to appropriate antimicrobial therapy.

- Thank you for this interesting approach for the future perspectives of this work which we have now added in the paragraph 4.4 New diagnostic techniques (previously 4.3) as following:

Besides the above mentioned and discussed scores, new diagnostic techniques try to approach the challenge of accurate and early diagnosis of NS. Machine learning is a subfield of artificial intelligence and uses particular electronic data to train and validate artificial Neural Networks in order to create diagnostic models and algorithms. Α great number of data (signs, symptoms, and especially monitoring and laboratory data) are gathered by septic and non septic infants and assemble a computer-based algorithm which, based on precise coefficients of variables, can diagnose or predict NS. The outcome of these models can be sepsis/no sepsis or/and a stratification of possibility for NS. The main purpose of this approach is to introduce a more personalized basis for diagnosis of neonatal sepsis, relying on precise and continuous information. A significant number of such studies reflect the interest on this new wrinkle, with impressive results in their diagnostic power [. Undoubtedly, this idea provides a pioneering perspective on the field, not only for the current times when NICUs count on continuous monitoring results, but also for the near future when computing methodologies will play a crucial role in medical decision-making generally. In our review, these studies were excluded because we did set a strict limit to studies containing arithmetical scores only. However, pre-existing data presented in this manuscript can reveal trends that are missed by cursory or even more detailed analysis. The use of high-speed computing where multiple factors are encorporated  could determine the optimal combination of clinical and laboratory based criteria. The reported sensitivity and specificity values in Table 4 along with the positive/negative predictive values could be used in this type of approach. Instead of a single score, a two tiered scoring system could be developed. One that integrates the intrinsic value of each defined measurement for early screening and the later for a definitive diagnosis. For instance, the early score is based on high sensitivity and for screening infants for NS using combined clinical and laboratory data suggested by computer analysis. Sensitivity is given priority over specificity when screening for NS so that false negatives could be avoided. However, this will lead to an increased level of false positives. Consequently, the definitive score would be used to rule out any false positives. This score would need to be highly specific but less sensitive. This approach should make the final determination of LOS more objective and lead to appropriate antimicrobial therapy.

Specific comments:

- Line 29 - Recommend additional characterization of neonatal sepsis indicating that it is a clinical syndrome that may include signs of systemic infection, circulatory shock, and multisystem organ failure.

We have added this comment in the revised version of the manuscript: The term ‘neonatal sepsis’ (NS) is used to describe the systemic condition caused by bacteria, viruses or fungi in the first four weeks of life and is a clinical syndrome that may include signs of systemic infection, circulatory shock, and multisystem organ failure

- Line 55 - The authors indicate microbe culture recovery as the “gold standard” but neglect mentioning rapid detection of certain neonatal sepsis agents such as Streptococcus agalactiae (group B streptococcus) using latex agglutination test (LAT) method suitable for use with either cerebrospinal fluid or blood samples.Line 58 - If antimicrobials are likely, blood culture bottles containing resin to neutralize antibiotics (e.g., BBL™ SEPTI-CHEK™) should be routinely included in suspected cases of neonatal sepsis to prevent false negative results.Line 63 - Consider mentioning that procalcitonin is being investigated as an acute-phase reactant marker for neonatal sepsis. It is very sensitive, when compared to C-reactive protein (CRP) but is less specific.

Following your three comments above we have rewritten the paragraph on current status of diagnosis of NS:

 Blood cultures may be false positive (due to contamination from the skin) or false negative (low volume of blood in neonates, previous antimicrobial therapy etc)[19-20]. It is worth mentioning at this point that if antimicrobials have been previously used, specific blood culture bottles containing resin to neutralize antibiotics (e.g.,BBLTM SEPTI-CHEKTM) should be used to prevent false negative results[17-20]. In addition, rapid detection of certain pathogens such us Streptococcus agalactiae using latex agglutination test (LAT) method may be suitable for use with either cerebrospinal fluid or blood samples[17-20]. Despite the above, the accurate diagnosis of NS remains challenging. Therefore, the term ’clinical sepsis’ (clinical features with negative cultures) is a well recognised entity by practising clinicians and may be more commonly encountered than positive-culture sepsis [[vii],[viii]]. Furthermore, another reason that makes detection of neonatal sepsis problematic is the relative diagnostic inaccuracy of the available parameters or biomarkers [17-24]. Procalcitonin appears to be more promising in the field with higher sensitivity compared to C-reactive protein(CRP). [17-24]

- Line 72 - The authors should consider outlining criteria included in Töllner’s scoring method and saying why it is inadequate.  Also, state major barrier(s) to developing an objective/accurate single scoring method.  

Following your comment we have added the following in the introduction section of the manuscript:

From 1974 to 1979, U. Töllner has made the first attempt to improve the early diagnosis of septicemia in newborns by the use of a score, which was published in 1982. In this first study, clinical and haematological findings of newborns  with NS were studied retrospectively with the view of creating a score which was then validated prospectively [24]. Since then, numerous studies have been published, reflecting the progressing interest in the field and the need for an accurate scoring system. Despite the remarkable efforts, a single score has not as yet been developed due to various barriers such as accurate and unanimous definition of neonatal sepsis, availability of laboratory tests and biomarkers in different resource settings, applicability and validation in different neonatal populations [9,10,12,13,15]. The aim of the current study is to assess the diagnostic value of predictive scores for LOS as a tool for early sepsis recognition as well as an antimicrobial stewardship tool.

- Line 174 - “The clinical score was calculated at 0h and at 24h of the onset of the illness”. Indicate that these time points are associated with early-onset (<72hours).

This sentence refers to the onset of illness rather than day of life

- Table 3 - Points (  ) missing after Abnormal WBC count (ref: Griffith et al 2007).

We have added these

- Table 4 - Specify source of Sensitivity/Specificity and Positive Predictive Value/Negative Predictive Value information (e.g. each study, calculated, etc.).

We have rephrased the title of Table 4 to accommodate the reviewer comment:

Diagnostic power of predictive scores as calculated in each study

- Line 295 - Use “full-term” instead of “terms” We have corrected this

Reviewer 3 Report

Neonatal sepsis and predictive scores is a  very challenging subject with a high amount of literature informations including clinical and laboratory parameters. The authors performed a systematic search revealing the most relevant predictive scores published from 1982 and discussed the elements of novelty and implication in clinical practice.

Author Response

We would like to thank the reviewer for the positive feedback on our manuscript

Reviewer 4 Report

My main concerns are the methodology and the way the result are reported.

·        The pico is not formulated correctly. Based on the way the results are presented I think the authors meant to do a PICO for diagnostic purposes: Population, Index test, Comparator test and test accuracy (sensitivity, NPV etc) as Outcome. Or, alternatively, a PICO for test accuracy: : P is the population of interest, I positive risk score, C negative risk score and O for the target disease (LOS).  If they want to look at the effect of the score itself (as an intervention) on patient outcome and antibiotic consumption (as the title suggests), the pico should be formulated as such (“normal” PICO with use of the score as intervention and patientoutcome and antibiotic consumtion as Outcome). But that does not seem the case.  

If the authors focus on accuracy of the test for predicting LOS (does the risk score predict LOS accurately enough) the PICO is a diagnostic one. In contrast, if the aim of the study is to look at the effect of risk scores on patient outcome and antibiotic use, the PICO is different, in that case the test itself is the intervention.

I think the the PICO, the methodology and the results should be aligned to fit the aim/clinical question. Now they do not.

·        The outcome of the pico is now defined as LOS, but the studies probably have different definitions of LOS, and these are not reported. This is essential information. As the authors discuss, defining LOS by culture results only is an underestimation. Therefore it is important to repot definition of LOS in the included studies

·        The search is limited to Pubmed, relevant studies could have been missed

·        Some studies report on both LOS and EOS, how is the inclusion of EOS handled when reporting the results? (i.e. 109

·        I suggest reconsidering the way the results are reported. In my opinion you need two tables only. One with the main results (table 2,3 and 4 combined): summarising, type of study, validated or not, outcome definition, components of score, and performance of score. Second you need a table reflecting the quality of the studies (now missing). Maybe discuss other relevant findings in the text.

·        The text in the results section could be improved, it describes with studies were done, but does not provide any information that supports the PICO.

·        Not all conclusions in the discussion section are supported by results. For example the authors report on the most important clinical symptoms for predicting LOS, but this is not supported enough by the results section  (r 336)

Minor

-        Not all abbreviatons are explained

-        The title does not reflect the result. stewardship.

-        The section on study selection is unclear (maybe adjust formulation) r 103-109

-        The relevance of dividing the existing scores (clinical, lab, risk, management, etcetera) is not clear to me. R 158. If it is kept this way, it should be explained. For example, in a risk score, all elements (including laboratory and clinical factors) are risk factors so it should be made clear what is meant by these categories.

-        What is meant by hematological symptoms (199)

-        The authors say a meta-analysis was included, but we did not find it in the results section.

-        The English needs editing
